# Different Concentrations of Probiotic *Pediococcus pentosaceus* GT001 on Growth Performance, Antioxidant Capacity, Immune Function, Intestinal Microflora and Histomorphology of Broiler Chickens

**DOI:** 10.3390/ani13233724

**Published:** 2023-12-01

**Authors:** Gifty Ziema Bumbie, Leonardo Abormegah, Peter Asiedu, Akua Durowaa Oduro-Owusu, Frederick Danso, Kwabena Owusu Ansah, Taha Mohamed Mohamed, Zhiru Tang

**Affiliations:** 1Laboratory for Bio-Feed and Molecular Nutrition, College of Animal Science and Technology, Southwest University, Chongqing 400715, China; giftyziema@gmail.com (G.Z.B.); tahaabdelameed@alexu.edu.eg (T.M.M.); 2Council for Scientific and Industrial Research, Animal Research Institute, Accra 20, Ghana; davinciabor@yahoo.com (L.A.); sikafuturoa111@gmail.com (A.D.O.-O.); ojake6@gmail.com (K.O.A.); 3Department of Animal Production and Health, School of Agricultural and Technology, University of Energy and Natural Resources, Sunyani 214, Ghana; pierro605@gmail.com; 4Council for Scientific and Industrial Research, Oil Palm Research Institute, Kade 74, Ghana; dansotodanso@gmail.com; 5Department of Animal and Fish Production, Faculty of Agriculture (Saba Basha), Alexandria University, Alexandria 21531, Egypt

**Keywords:** *Pediococcus pentosaceus* GT001, probiotics, antibiotics, broilers, histomorphology

## Abstract

**Simple Summary:**

Because broiler chickens can meet human nutritional demands, they are commonly consumed in many nations worldwide. Although the abuse of antibiotics in the chicken business has had major negative effects on public health, antibiotics have been used in broiler diets to minimize poultry infections and increase growth performance. Probiotics are therefore a secure and healthful substitute for antibiotics. *Pediococcus pentosaceus* is a promising strain of lactic acid bacteria (LAB) which is gradually attracting attention, leading to a rapid increase in experimental research. *Pediococcus pentosaceus* strains can be applied as an animal growth bio-promoter and as a probiotic. However, information on the use of *Pediococcus pentosaceus* as a probiotic remain scanty. This study was therefore conducted to investigate the effects of different doses of *Pediococcus pentosaceus* GT001 supplementation on the growth performance, immune function, intestinal development and histomorphology of broilers and to understand the protective mechanisms of *Pediococcus pentosaceus* GT001 probiotics on the host.

**Abstract:**

Exploring alternatives to antibiotics is imperative in reducing antibiotic resistance and antibiotic residues in poultry products. The beneficial effects of antibiotic products derived from natural sources in comparison with the synthetic ones has been reported. *Pediococcus pentosaceus* has been applied as an animal growth bio-promoter and probiotic. To elucidate the protective mechanisms of *P. pentosaceus*, this study investigated the effects of different doses of *P. pentosaceus* supplementation on broiler growth performance, immune function, intestinal development and histomorphology. Five hundred (500) one-day-old Ross 708 broiler chicks were randomly enrolled into five experimental groups with 20 chicks per replicate. The treatments were imposed as follows: (T1) basal diet (control); (T2) basal diet with 1 g/kg antibact 3X; (T3) basal diet with *P. pentosaceus* GT001 at 4.0 × 10^8^ cfu/g; (T4) basal diet with *P. pentosaceus* GT001 at 8.0 × 10^8^ cfu/g; and (T5) basal diet with *P. pentosaceus* GT001 at 1.2 × 10^9^ cfu/g. Dietary inclusion of *P. pentosaceus* GT001 at 4.0 × 10^8^ cfu/g significantly improved body weight gain, feed intake and lipid profile of the broilers compared to the control group (*p <* 0.05). The addition of *P. pentosaceus* GT001 significantly improved the intestinal pH of the broilers. The digestive enzymes of the broilers were impacted with the supplementation of *P. pentosaceus* GT001 at 4.0 × 10^8^ cfu/g. The highest serum antioxidant production was observed in the *P. pentosaceus*-treated group compared to the control. *P. pentosaceus* GT001 at 4.0 × 10^8^ cfu/g increased the levels of serum cytokines and immunoglobin and improved the small intestinal morphology of the broilers in comparison with the control. The load of *Pedococcus spp* was similar among T3, T4 and T5 but significantly higher than that of the control (T1) and the antibiotics (T2)-fed birds. The load of *E. coli* in the gut was significantly reduced in T3, T4 and T5 compared to T1 and T2. There was no *Salmonella* growth among the treatments. This study highlights the importance of probiotics in broiler diets and suggests that *Pediococcus pentosaceus* GT001 could be used as a feasible substitute to antimicrobials in broiler production.

## 1. Introduction

The multi-billion-dollar poultry trade attributed to the continuous demand for the product requires high efficiency in production and high stocking densities. This subsequently exposes poultry to stressful conditions resulting in low growth rates, disease and death [1]. In intensive commercial poultry production, antibiotics are used at sub-therapeutic doses in order to prevent diseases and improve productivity [2]. However, antibiotic exploitation over the years has led to a reduction in animal performance and feed conversion efficiency and has increased the prevalence of poultry diseases. Furthermore, with the increasing awareness of food safety, the problem of antibiotic residues in animal products has become a focus of attention. Therefore, exploring alternatives to antibiotics is imperative in reducing antibiotic resistance and antibiotic residues in poultry products. In recent years, some studies have reported the beneficial effects of antibiotic products derived from natural sources in comparison with the synthetic ones [3]. Probiotics are a group of microorganisms that are beneficial to the health of the host [4]. The supplementation of probiotics has exhibited evidence of an effective natural method for controlling animal gut flora. It can maintain the ecosystem of gut microflora, prevent the growth of pathogens, boost the activity of endogenous digestive enzymes and function as a positive immune modulator by preserving the integrity of the intestine [5,6] and the immune response of the host [7]. Probiotic supplementation has a significant effect on carcass yield, live weight gain, immune response and prominent cut-up meat parts [8]. Therefore, strategically using probiotics in broiler diets can enhance the synthesis of antimicrobial agents, immunomodulation, competition for adhesion sites, diversity and stability of the intestinal microbiota, all of which lead to better broiler performance [9]. Experimental study of Pediococcus pentosaceus, a promising strain of lactic acid bacteria (LAB), is rapidly increasing as it gains more attention [10]. Certain *P. pentosaceus* strains have been shown to be effective as probiotics and as a bio-promoter of animal growth since the 1990s [11]. Nevertheless, there is still a dearth of knowledge regarding *P. pentosaceus*’s usage as a probiotic. Furthermore, little is known about the mechanisms, adverse effects, application and dose of *P. pentosaceus* and its bacteriocins in the production of broilers, despite mounting evidence to the contrary.

To further clarify the probiotic properties of *P. pentosaceus* on broiler production, this study was conducted to investigate the effects of different doses of *P. pentosaceus* GT001 supplementation on the growth performance, immune function, intestinal development and histomorphology of broilers and to understand the protective mechanisms of *P. pentosaceus* probiotics on the host. These results will give helpful information on the effects and application of *P. pentosaceus* probiotics in broiler production. 

## 2. Materials and Methods

### 2.1. Ethical Approval

The probiotic isolation and identification and the birds’ care and protocol used were approved by the Council for Scientific and Industrial Research (CSIR) Institutional Animal Care and Use Committee (IACUC), Ghana. The research complied with the protocol’s requirements RPN 008/CSIR-IACUC/2022 approved by the IACUC ethics committee (approval date: 21 July 2023).

### 2.2. Pediococcus Pentosaceus Production and Administration

The *Pediococcus pentosaceus* GT001 used in this experiment was isolated, cultured and tested in vitro in a previous study (as part of PhD work). A fresh culture of the *Pediococcus pentosaceus* GT001 probiotics was resuscitated and inoculated in MRS broth medium overnight at 37 °C; the overnight cultures were centrifuged at 3000 rpm for 15 min, washed twice with sterilized phosphate-buffered saline (PBS) (pH 7.4) and resuspended in PBS to adjust the concentration to 4.0 × 10^8^ CFU/g, 8.0 × 10^8^ CFU/g and 1.2 × 10^9^ CFU/g. Approximately, 10 mls of the *Pediococcus pentosaceus* GT001 probiotics with adjusted concentrations was mixed thoroughly with 100 g of feed. 

### 2.3. Birds, Housing, Diet and Experimental Design 

The study area was cleaned and disinfected prior to receiving the birds. Five hundred (500) healthy one-day-old mixed-sex Ross 708 broiler chicks of average body weight (BW) of 38.34 g at day one were obtained from Pluriton, Belgium. The chicks were randomly allotted to a total of 25 floor pens (area of 3 m × 2.25 m) covered with fresh wood shavings. The birds were raised in a deep litter, with each pen having one (1) feeder, one (1) drinker and stress-free access to feed and water ad libitum during the course of the experiment. The experimental diet was based on maize and soybean meal and designed according to the NRC [12] requirements for both the starter (1–21 days) and finisher period (22–42 days). The feed was in a mash form. All birds were fed the basal diet throughout (day 1 to 42) the experiment. The composition and nutrient levels of the basal diet are shown in Table 1. There were five experimental treatment groups and each consisted of five replicates, with 20 chicks per replicate, in a completely randomized block design. The treatments were as follows: (T1) basal diet (control); (T2) basal diet with 1 g/kg antibact 3X; (T3) basal diet with *P. pentosaceus* GT001 at 4.0 × 10^8^ cfu/g; (T4) basal diet with *P. pentosaceus* GT001 at 8.0 × 10^8^ cfu/g; and (T5) basal diet with *P. pentosaceus* GT001 at 1.2 × 10^9^ cfu/g. Approximately, 10 mls of the probiotics was added to 100 g of the basal diet, mixed thoroughly and fed to the birds after an hour of withdrawal from feed, while the antibiotic was in powder form and added to the basal ration according to the experimental design. A total of 1 kg of ANTIBACT 3X contains the active ingredients tylosin tartrate, Oxytetracycline HCl and Neomysin Sulphate.

### 2.4. Management

Each pen was covered with clean wood shavings which were changed as and when necessary. Using controlled heaters, fans and the opening of doors and windows, room temperature was maintained at 34 °C for the first 5 days and then gradually reduced to 22 °C till the end of the experiment, according to the standard management practices. Chicks in all treatment groups were vaccinated against Newcastle disease (YEBIO^®^) using LaSota B1 Strain of Newcastle disease virus in live freeze-dried form and IBD against Gumboro disease by adding it to their water at day 14 and 21, respectively. The study area was routinely cleaned and disinfected to prevent the occurrence of diseases. 

### 2.5. Growth Performance

Daily feed intake per replicate was recorded to compute feed intake per week (day 7, 14, 21, 28, 35 and 42). Growth performance (body weight, body weight gain, feed intake and feed conversion ratio) was measured per replicate unit from 1–42 days of age. Values of feed intake and weight gain were used to calculate the feed conversion ratio. Mortality and mortality rate per treatment were recorded and calculated for the experimental period.

### 2.6. Serum Samples

Birds with similar average body weight were chosen from each treatment replicate pen at the end of the experiment (day 42). Five mL blood samples from the jugular vein were collected from two birds per replicate in each treatment into vacutainer tubes and allowed to clot at room temperature. The clotted blood samples were centrifuged at 3000 rpm for 15 min at room temperature to separate the serum from the blood cells. The harvested serum samples were stored at −20 °C for further analysis.

#### Biochemistry Analysis

The Biobase BK-200 mini Automated Chemistry Analyzer was used to determine the biochemistry analysis. Total protein, albumin, globulin, creatinine, aspartate aminotransferase (AST), alanine aminotransferase (ALT), cholesterol, triglyceride, high density lipoprotein (HDL), low density lipoprotein (LDL), total antioxidant capacity (T-AOC), superoxide dismutase (SOD) and glutathione peroxidase (GSH-Px) activities were measured from the stored serum samples. Additionally, malondialdehyde (MDA), interleukin-10 (IL-10), interleukin-6 (IL-6), tumor necrosis factor-a (TNF-a), immunoglobulin A (IgA), immunoglobulin G (IgG) and immunoglobulin M were assessed. Utilizing ELISA kits designed specifically for chickens, the concentrations were found. The technique followed the manufacturer’s protocol, and the ELISA kits were purchased from Nanjing Jiancheng Bioengineering Institute (Nanjing, China).

### 2.7. Intestinal Measurement and pH Determination 

From the selected slaughtered birds (two birds per replicate) at day 42, the intestinal content from the duodenum, ileum and jejunum were collected into sterile plastic containers and a pH probe (SP-701/pH/mV/Temp.Meter, Suntex, Taipei, Taiwan) was placed directly into the digesta content to record the pH. The duodenal, ileal and jejunal lengths were measured in cm using a measuring tape. 

### 2.8. Digestive Enzymes

Serum digestive enzymes amylase and lipase were determined using ELISA kits acquired from Nanjing Jiancheng Bioengineering Institute (Nanjing, China) and the procedure used was based on the manufacturer’s protocol. With the intestinal digestive enzymes, the small intestinal digesta were homogenized with 0.9% physiological saline, and then the homogenate was centrifuged at 5000× *g* for 15 min, and the supernatant was collected. Amylase and lipase activities in the small intestinal digesta were then measured with detection kits from Nanjing Jiancheng. 

### 2.9. Intestinal Histology and Morphology 

Segments from the midpoint of the duodenum, jejunum and ileum were collected at day 42 from the selected slaughtered birds, flushed with a 0.9% salt solution and fixed in 10% formaldehyde–phosphate buffer for 48 h. Under a light microscope, the sections were then stained with hematoxylin–eosin, and the depth of the intestinal crypts as well as the height and width of the intestinal villi were measured. For every intestinal cross-section, ten complete, correctly aligned crypt–villus units were chosen in triplicate. The ratio of crypt depth to villus height was computed. A Leica DM500 light microscope that was connected to a Leica Microsystem Framework integrated digital imaging analysis system (Leica ICCSO HD, Heerbrugg, Switzerland) was used to investigate histological segments.

Villi height was vertically measured from the villi–crypt junction to the tip of villi, whereas crypt depth was measured from the root of the lower limit of the crypt to the villi–crypt junction [13]. 

### 2.10. Intestinal Microflora

Samples of the digesta content from the duodenum, ileum and jejunum were aseptically collected from the selected slaughtered birds into sterile plastic containers and transported to the laboratory in liquid nitrogen. Digesta samples were kept in −40 °C until the analysis of microbial count. In the lab, 1 g of digesta samples from the 3 sections was diluted with 9 mL of peptone water and 10-fold serially diluted. Diluted samples (0.1 mL) were inoculated into selective agar, and further bacterial enumeration was determined in a biosafety cabinet. *Salmonella* was incubated using XLD agar; *E. coli* and *Enterococcus* were incubated using Chromogenic UTI Medium. *Enterobacter*, non-lactose fermenter, total coliform, total viable count and *Pediococcus pentosaceous* were incubated using MacConkey’s t1905 crystal violet bile salts neutral red agar, MacConkey agar, VRBA, Standard Plate Count agar and MRS agar, respectively. The population of microbes was expressed as log_10_ colony-forming units/g of the digesta.

### 2.11. Statistical Analysis

All data are presented as mean ± SEM. Data were analyzed as one-way ANOVA using the general linear model (GLM) of Minitab^®^ version 18.1 (Minitab version 18) as a randomized complete block design (RCBD) with five replications. The test for differences among treatment means was conducted using Tukey’s test, and statistical significance was assumed at *p* < 0.05. 

## 3. Results

### 3.1. Growth Performance 

The results in Table 2 show the effect of different concentrations of *Pediococcus pentosaceus* GT001 on the growth performance of broiler chicken. The initial weight of the birds prior to the commencement of the experiment was similar among the dietary treatments. At the end of the experiment, the final bird weight increased with dietary treatments in all treatments. The final weight of the birds in both T1 and T3 were significantly higher compared to T2 and T5 (*p* = 0.030) but were similar to T4. The feed intake day^−1^ of T3 was the highest and varied significantly from the other treatments (*p* = 0.001), while T2 recorded the lowest feed intake day^−1^ during the period of the study. The adoption of T4 was significantly higher than both T5 and T2 in terms of feed intake day^−1^ (*p* = 0.001) but was similar to T1. The responses of T1 and T3 were higher and differed significantly from both T2 and T5 in terms of the total weight gain (*p* = 0.030) but were similar to T4. Similarly, the ADG of T1 and T3 were higher and significantly different from both T2 and T5 (*p* = 0.030). The FCR values showed no significant responses (*p* = 0.514) to any of the treatments imposed on the birds.

### 3.2. Liver Function 

The results of the different concentrations of *Pediococcus pentosaceus* GT001 on the liver function of the broilers is shown in Table 3. The total protein of the treatments did not vary significantly (*p* = 0.970). Similarly, the albumin and globulin content showed no significant variation among any of the treatments imposed. The creatinine content of T1 did vary significantly from T2, T3 and T4 (*p* = 0.003). However, the different *Pediococcus pentosaceus* GT001 levels in the broiler diet did not significantly influence the creatinine content during the study. Among the treatments, the AST content did not vary significantly (*p* = 0.580). The T2 and T5 groups showed significant variation from T1, T3 and T4 in terms of ALT content.

### 3.3. Lipid Profile

The lipid profile of the broilers is shown in Table 4. Compared to T1, T2 and T5 varied significantly in terms of the cholesterol content. Increasing the *Pediococcus pentosaceus* GT001 from 4.0 × 10^8^ cfu/g feed to 8.0 × 10^8^ cfu/g feed did not cause the cholesterol content to vary significantly (*p* = 0.001). Similarly, the cholesterol content did not vary significantly when *Pediococcus pentosaceus* GT001 was increased from 8.0 × 10^8^ cfu/g feed to 1.2 × 10^9^ cfu/g feed. The triglycerides content of T3 varied significantly from both T2 and T4 (*p* = 0.001). The HDL content of T3 was significantly lower compared to T4 (*p* = 0.040). The choice of both *Pediococcus pentosaceus* GT001 at 4.0 × 10^8^ cfu/g feed and at 8.0 × 10^8^ cfu/g feed showed significantly higher LDL content as compared to T5 and T2 (*p* = 0.001).

### 3.4. Intestinal pH and Length

The pH of the duodenum showed no significant differences among the treatments (*p* = 0.879), as shown in Table 5. The jejunum pH of T1 and T2 was similar but both were higher than and significantly different from both T3 and T4 (*p* = 0.046). T1 was similar to T2 in the pH of the ileum but no significant difference was noted between T1 and T3, T4 and T5 (*p* = 0.028). The duodenum length of the different treatments did not vary significantly (*p* = 0.300), as shown in Table 5. The longest duodenum length was produced in T5, while the lowest duodenum length was recorded in T1 during the period of the study. The longest jejunum length was noted in T5, while the shortest length was produced by T3. Regarding the ileum length, T1 recorded the highest, while the lowest was noted in T5.

### 3.5. Serum and Intestinal Digestive Enzymes

The results of the digestive enzymes are shown in Table 6. The serum amylase of T3 and T4 recorded the highest and most significant values compared to T1, T2 and T5 (*p* = 0.001). The serum amylase of T1 and T5 varied significantly from T2 (*p* = 0.001). The adoption of T3 produced the highest and most significant serum lipase compared to the other treatments (*p* = 0.001). While no significant variation was noted in the serum lipase of T1 and T2, significant differences in the serum lipase were observed between T4 and T5 (*p* = 0.001). The use of T2 produced the lowest and least significant value of intestinal amylase compared to the other treatments (*p* = 0.001), although T1, T3, T4 and T5 recorded no significant differences among them (*p* = 0.001). Of the treatments in the study, the usage of T2 produced the lowest and least significant value of intestinal amylase compared to the other treatments (*p* = 0.001). The intestinal lipase of T3 recorded the highest and most significant value compared to T1, T2, T4 and T5 (*p* = 0.001). The use of T1 and T3 differed significantly in terms of intestinal lipase, although T4 and T5 did not vary significantly (*p* = 0.001) during the period of study. 

### 3.6. Antioxidant Capacity

Table 7 shows the serum antioxidant capacity during the study period. The T-AOC varied significantly among the dietary treatments, with T3 recording the highest and most significant value (*p* = 0.001). The use of T1 and T2 produced a significantly different T-AOC, while it did not vary significantly among T2, T4 and T5 (*p* = 0.001). Significant variations in SOD values were noted between T4 and T1, T2 and T5, with the exception of T3 (*p* = 0.001). T5 did vary significantly from T1, T2 and T4, with the exception of T3. The adoption of T1 showed significant differences from T2. The GHS-Px of T4 recorded the highest significant differences from T1, T2 and T3, with the exception of T5 (*p* = 0.001). Although both T4 and T5 exhibited no significant differences, both differed significantly from T1 and T2 (*p* = 0.001). The CAT values of T3, T4 and T5 showed no significant differences, although they all varied significantly from T1 and T2 (*p* = 0.001). The adoption of T2 differed significantly from T1 (*p* = 0.001). The MDA of T1 and T4 recorded the highest and lowest significant values, respectively (*p* = 0.001). T4 and T5 showed no significant differences.

### 3.7. Serum Cytokines and Immunoglobin 

The cytokines in the serum are represented in Table 8. Significant variation was observed among the different treatments of the study. T1 differed significantly from T2, T3, T4 and T5, recording the highest TNF-α value. The use of T4 produced the lowest significant difference of the TNF-α value (*p* = 0.001), while T2 was similar to T3. The IL 6 of T1 and T3 were significantly different from T2, T4 and T5 (*p* = 0.001). Both T2 and T4 recorded the lowest IL 6 values and differed significantly from T5 (*p* = 0.001). The IL 10 values of T3, T4 and T5 were significantly higher than those of T1 and T2 during the period of the study. There was a difference between T1 and T2 in terms of IL10. Table 8 shows the immunoglobin values recorded during the study. The IgA values of T3 showed significant variation from T1, T2 and T5 (*p* = 0.001). Both IgG and IgM values of T3, T4 and T5 differed significantly from T1 and T5 (*p* = 0.001).

### 3.8. Intestinal Morphology

The small intestinal morphology of the birds is shown in Table 9. The T3 treatment recorded significant differences in duodenum villus height from T1, T2 and T5 (*p* = 0.001). Additionally, both T1 and T5 differed significantly from T2 (*p* = 0.001). In terms of duodenum crypt depth, only T4 and T5 differed significantly from each other (*p* = 0.023). The duodenum VH/CP ratio showed significant variation between T2 and both T3 and T5 (*p* = 0.010). The ileum villus height of T1, T3 and T5 only varied significantly from T2 during the study (*p* = 0.018). T1 showed the highest significant ileum crypt depth difference among the treatments imposed (*p* = 0.001). Although the use of T2, T4 and T5 did not show any significant variation in the ileum crypt depth, they all differed significantly from T3 (*p* = 0.001). The ileum VH/CP ratio of T3 was the only treatment that exhibited the highest significant difference among the treatments imposed during the study (*p* = 0.001). The ileum VH/CP ratio was similar among the dietary treatments of T1, T2, T4 and T5. The jejunum villus height varied significantly only between T3 and T1 (*p* = 0.018). Among the treatments, only T3 varied significantly from both T1 and T5 in terms of jejunum crypt depth (*p* = 0.002). T3 recorded the highest significant VH/CP ratio value among the treatments, while T1 and T5 differed significantly from each other during the study (*p* = 0.001).

The Figure 1 is showing electronic images of the different sections of the small intestine among the dietary treatments tested.

### 3.9. Intestinal Microflora Count

The microbial load results obtained from the duodenum content are shown in Table 10. The load of *Pedococcus spp* was similar among T3, T4 and T5 but significantly higher than that of the control (T1) and the antibiotic (T2)-fed birds. The total viable count of T1 and T2 recorded the highest significant values compared to the different concentrations of the *Pediococcus pentosaceus* GT001 (T3, T4 and T5). No significant difference was recorded for *E. coli* among the dietary treatment. In terms of other microbial analyses such as *Enterobacter, Enterococcus, Salmonella* and non-lactose fermenter count, no growth was recorded for any of the treatments, as shown in Table 10.

The microbial load results obtained from the jejunum of experimental broiler chickens are presented in Table 11. A significant difference was recorded in the *Pediococcus* spp. load. The T3, T4 and T5 treatments were similar, but these treatments were significantly higher than the T1 and T2 treatments. In terms of the total viable count, there was a significant difference among the treatments. There was a significant variation in the total coliform count in the jejunum. The T1 and T2 treatments were similar but significantly higher than the treatments T3, T4 and T5. Additionally, T5 was significantly lower compared to T4 and T5. The *E. coli* count was similar among T2, T3, T4 and T5, but these treatments were significantly lower than T1. There was a significant variation in the *Enterococcus* count as well; T5 recorded the lowest and T2 recorded the highest. The other microbial counts such as non-lactose fermenters, *Enterobacter* and *Salmonella* recorded no growth.

The microbial load results obtained from the ileum of the experimental broiler chickens are shown in Table 12. All of the microbes counted in the ileum recorded growth, except from *Salmonella.* Significant variation was observed among the different treatments of the study. The *Pedococcus* spp. count was similar between T4 and T5 but significantly higher than that of T3. Also, T1 and T2 were similar but significantly lower than T3. In terms of the total viable count, the control recorded the highest value and it was significantly higher than the other treatments; T2, T3, T4 and T5 were similar. There was no significant difference between the birds that were fed the different concentrations of the *Pediococcus pentosaceus* GT001 (T3, T4 and T5) when it comes to the total coliform count, but the values obtained were significantly lower than the control birds (T1). Additionally, T1 and T2 were similar. No significant variations were observed among the treatments for non-lactose fermenters and *Enterobacter* count. T5 recorded the lowest *E. coli* count and it was significantly lower compared to the other treatments. T1 and T2 were similar but significantly higher than T3 and T5. There was a significant variation in the values recorded for *Enterococcus.*

## 4. Discussion

### 4.1. Growth Performance

The selection of safe and appropriate antibiotic alternatives that give economic returns is essential in poultry production [14]. Probiotics can hinder host infection by pathogens and potentially replace antibiotics. This study observed significant increases in the final weight, feed intake, total weight gain and average daily weight gain of the birds that received the basal diet supplemented with *P. pentosaceus* GT001 at 4.0 × 10^8^ cfu/g feed and *P. pentosaceus* GT001 at 8.0 × 10^8^ cfu/g feed. The beneficial effect of *P. pentosaceus* GT001 on growth performance is attributable to the improved feed digestibility and the action of beneficial bacteria in the gut [15]. A higher concentration of *P. pentosaceus* GT001 at 1.2 × 10^9^ cfu/g feed showed a reduction in the growth performance of the birds. This suggests that an adequate concentration of *P. pentosaceus* GT001 supplementation in poultry diets would provide a favorable environment which assists microflora colonization in the intestine for better growth performance of birds [16].

Genetic factors play a significant role in feed conversion efficiency in Ross 708 broilers. These factors influence the bird’s ability to metabolize and convert feed into body weight gain, ultimately affecting their overall performance [17]. Environmental factors such as temperature, humidity, lighting and air quality also play a significant role in affecting feed conversion in Ross 708 broilers. Temperature can greatly impact feed conversion, as broilers have a narrow temperature range in which they perform optimally. High temperatures can increase water consumption, leading to higher feed conversion ratios. Humidity also affects feed conversion, with higher humidity reducing the birds’ ability to dissipate heat and potentially increasing stress levels [18].

### 4.2. Liver Function

Pietras et al. [19] reported a higher protein content of chickens given probiotics, contrary to the observation from our study. The total protein content did not vary significantly among the treatments imposed. The standardized basal diet may play a major role in the non-variation in the total protein among the birds. Alterations in the activities of AST and ALT are also specific indicators that can be utilized to ascertain the organism’s hepatocyte activity as well as specific indicators of hepatocyte damage [20]. As a result of increased cell membrane permeability brought on by hepatocyte injury, AST and ALT are released from the cytoplasm of the hepatocytes. Research conducted by Fathi [21] indicated that feeding chickens a diet containing Lactobacillus cultures greatly lowered their ALT and AST activity. In the current study, the supplementation of *P. pentosaceus* GT001 to the diet of broiler chickens did not significantly affect plasma AST levels, but ALT was significantly lower in T5 and the values obtained were within normal range. Maintaining AST and ALT levels within the range during the research experiment suggests that broilers’ liver function was normal when fed *P. pentosaceus* GT001. Creatinine levels in the plasma was significantly reduced by administering *P. pentosaceus* GT001. In consistent with the current research, Mohamed et al. [22] reported lower creatinine levels when the broiler diet was supplemented with probiotics.

### 4.3. Lipid Profile

*P. pentosaceus* have been shown to affect cholesterol metabolism by decreasing cholesterol levels [23]. From our study, lower cholesterol levels were noted with higher levels of *P. pentosaceus* GT001 at 1.2 × 10^9^ cfu/g feed compared to lower levels of either *P. pentosaceus* GT001 at 4.0 × 10^8^ cfu/g feed or *P. pentosaceus* GT001 at 8.0 × 10^8^ cfu/g feed. This indicates that higher levels of *P. pentosaceus* GT001 may be a precursor to lowering chicken cholesterol levels. Dietary supplementation with probiotics at lower doses lowered the triglyceride concentrations in the serum of broiler chickens in the study of Panda et al. [24]. Similarly, our study showed lower triglyceride levels in the birds that were fed doses of *P. pentosaceus* GT001 at 4.0 × 10^8^ cfu/g feed compared to either 8.0 × 10^8^ cfu/g feed or 1.2 × 10^9^ cfu/g feed.

The triglyceride levels in the experimental groups (T3 and T5) were lower, indicating the positive effects of probiotic supplementation on lipid metabolism. These findings suggest that probiotics play a crucial role in regulating triglyceride levels in broiler chickens, potentially leading to improved health outcomes [25]. HDL regulates cholesterol levels to prevent its accumulation in cells, and sterols are shed from membranes at the same rate at which cholesterol is synthesized in the liver to maintain a balance [26]. The function of HDL is to transport any remaining cholesterol that is not being utilized to the liver. The remaining cholesterol will be used as a component in the production of steroid hormones and bile salt, while the remaining inactive cholesterol will be excreted [27]. In our study, HDL levels of the probiotic-fed birds were low, while LDL levels were high. In contrast to our study, Mohamed et al. [22] reported high HDL levels and low LDL content in probiotic-fed broiler chicken.

### 4.4. Intestinal pH and Length

A number of reasons have been put forward to explain the beneficial effects of probiotics in lowering the pH of the gut, leading to the low stability of pathogenic bacteria in the intestines [28]. This was evident in the lower pH in the duodenum, jejunum and ileum of the birds that received either the basal diet supplemented with *P. pentosaceus* GT001 at 4.0 × 10^8^ cfu/g feed or *P. pentosaceus* GT001 at 8.0 × 10^8^ cfu/g feed. At higher quantities, *P. pentosaceus* GT001 might have stimulated antibacterial properties that are more pronounced at a lower pH, which may help reduce the bacterial burden or alter the distribution of bacterial species in the gut. According to other studies, *Lactobacillus* probiotics are capable of producing secondary metabolites including bacteriostatic toxin, releasing organic acid that reduces intestinal pH, and stopping the growth of dangerous bacteria to preserve the intestinal biological barrier [29]. The *Lactobacillaceae* family member *P. pentosaceus* may have a similar resistance function [30].

### 4.5. Digestive Enzymes

The positive outcomes of probiotics can be attributed to their numerous advantageous traits, and they have been suggested as a feasible substitute for antibiotics, particularly considering the current unregulated usage of antibiotics [31]. Particularly during development, the intestinal enzyme activity of the birds is crucial to the digestion and absorption of nutrients [32]. Probiotics have been shown by Wang et al. [33] to considerably raise the intestinal lipase activity of broiler chickens when added to their diet.

This was evident in the basal diet supplemented with *P. pentosaceus* GT001 at 4.0 × 10^8^ cfu/g feed in our study. Our study also found that adding *P. pentosaceus* GT001 at 4.0 × 10^8^ cfu/g feed into the diet significantly increased both the serum lipase and amylase activities of the birds, which t is inconsistent with previous studies [33]. Probiotics may stimulate endogenous enzyme synthesis in the intestines and secrete their products, which could account for the increase in digestive enzyme activity [34]. Furthermore, by maintaining intestinal integrity and morphology, dietary probiotics could boost levels of enzymes [35].

This indicates that the addition of probiotics in chicken diets effectively increases intestinal enzyme activities and improves nutrient uptake by broiler chickens [32].

### 4.6. Serum Antioxidant Capacity of Broiler Chickens

Antioxidant capacity is considered an important indicator of the bird’s immune function and for evaluating the oxidative status of animals [32,36]. The levels of MDA, SOD, GSH-Px and T-AOC are all important indicators which correlate to antioxidant capacities [27]. The addition of *P. pentosaceus* GT001 to the diet of the birds caused a significant increase in SOD, GHS-Px and CAT activities, as seen in the results obtained in this study. The results are in agreement with Zhang et al. [37], who reported that probiotics in broiler diets improved significantly SOD, GHS-Px and CAT activities in serum. Probiotics may also produce antioxidant metabolites that have strong antioxidant potential, such as antioxidant peptides [38], and they may also activate the Nrf2/Keap1 signaling pathway, which further increases the expression of the genes and activity of the antioxidant enzyme [39]. One important sign of oxidative stress is MDA, which is a lipid peroxidation indicator [40]. In this study, the addition of *P. pentosaceus* GT001 above 4.0 × 10^8^ cfu/g feed significantly reduced serum MDA levels in the birds, similar to the observations in the study by Wang et al. [41]. According to Yang et al. [42], this may occur due to improved SOD activity in HepG2 cells relieving the oxidative stress induced by H_2_O_2_. Our study also showed significant reductions in T-AOC levels in T4 and T5, and this could be attributed to the result of increased activities of antioxidant enzymes which may not be sufficient to counteract the oxidative stress caused by probiotic supplementation. This could be due to an imbalance between the production of reactive oxygen species and the antioxidant defense system [43]

### 4.7. Serum Cytokines and Immunoglobin

Systemic immunomodulatory effects are considered to be an important mechanism of probiotic function, and supplementation with probiotics has been reported to recruit immune cells, therefore activating immune or inflammatory responses by altering the synthesis of cytokines [44]. The difference in the cytokine concentration in the birds of our study is attributable to the different concentrations of *P. pentosaceus* GT001 added to the feeding regime. Studies by Zhang et al. [45] and Rajput et al. [46] showed that dietary supplementation with probiotics changed the sensitization of the host by increasing the concentrations of TNF-α, IL 6 and IL 10. In the present study, serum TNF-α, IL 6 and IL 10 were increased in the birds that were fed with *P. pentosaceus* GT001 at 4.0 × 10^8^ cfu/g. This result is in agreement with Selvam et al. [47], who observed that the inclusion of probiotics had significantly higher levels of anti-inflammatory cytokines in the serum. The immune response is a complex process involving the innate immune system, whose activation is indicated by the release of inflammatory factors, such as TNF-α, IL 6 and IL 10, which play an important role in this process [30]. Adding probiotics to the diet of birds has been demonstrated to help enhance their immune function, cellular immunity and humoral functions [48]. The primary components of the intestinal immune barrier are immunoglobulin and immunoactive cytokines, which are expressed as intestinal mucosa lymphocytes and support the intestinal tract’s local immunological activity [49]. Intestinal epithelium’s first line of defense against infection is secretory IgA, which keeps the gut’s equilibrium intact [50].

The addition of *P. pentosaceus* GT001 to the diet of the birds improved the IgA, IgG and IgM levels in this study. The results agree with Bai et al. [51], who reported that probiotics increased the serum IgA and IgG concentrations of broiler chickens. According to Peng et al. [32], the immune capacity of the body can be reflected by the level of these immunoglobulins in plasma. This indicates that *P. pentosaceus* GT001 is important in raising the immunoglobulins in plasma and building the immune responses of the birds. This may occur due to the presumption that probiotics need to consume a lot of free oxygen to multiply in the intestines, which improves the growth and multiplication of anaerobic probiotics and inhibits pathogen colonization [52].

### 4.8. Small Intestinal Morphology

Villus height and crypt depth are indicative of intestinal digestion and absorption function, as well as cell maturity rate, respectively [53]. The duodenum and ileum villus height, crypt depth and VH/CP ratio of the birds in our study that received *P. pentosaceus* GT001 at 4.0 × 10^8^ cfu/g feed or *P. pentosaceus* GT001 at 8.0 × 10^8^ cfu/g feed showed significant responses. This is in agreement with the study of Lee et al. [54] which showed that a diet supplemented with probiotics can promote the growth of intestinal epithelial cells, increase villus height of the small intestine and improve the absorption of nutrients. The higher VH/CP ratio in our study supports the assertion of some previous findings which suggested the beneficial effect of probiotics in providing an intestinal environment conducive to digestion, absorption of nutrients and intestinal health [55,56]. Therefore, *P. pentosaceus* GT001 may act as an effective alternative to antibiotics in poultry production.

### 4.9. Microbial Count

The intestinal microbiota and the host interact naturally through the gut. The gut microbiota is made up of millions of different genes and may consist of countless numbers of bacteria compared to the host’s cells. Gut microbiota have the ability to start several enzymatic reactions that the host is unable to carry out by expressing these special genes. As a result, the gut microbiota has a significant impact on the development of the gut and the host’s metabolism [57]. Generally, it is accepted that probiotics’ ability to promote growth is a result of the development of the gut microbiota and their role in beneficial processes in the intestine. Also, by limiting the spread of pathogenic species and boosting the population of helpful bacteria, probiotics in the diet help healthy hosts maintain a healthy balance of their gut microbiota. The inclusion of *Pediococcus pentosaceus* GT001 as probiotic in the diet of broiler chickens significantly decreased the *E. coli* count and with no growth of *Salmonella* in this study. According to Salim et al. [58], adding a probiotic to chicken diets had no effect on the amount of *Salmonella* present. The *E. coli* count was, however, drastically declined in birds administered probiotics. Yang et al. [59] also reported a reduction in the levels of *Salmonella* and *E. coli* in broiler cecal contents when a probiotic was added to the diet. Probiotics have been shown to affect the gut microflora through a variety of mechanisms including competitive exclusion, acid fermentation, which lowers pH, increased production of short-chain fatty acids, competition for nutrients and mucosal sites of attachment, stimulation of the gut’s associated immune system and increased epithelial integrity [58]. No beneficial consequences were observed on cecal bacteria counts when the birds were fed probiotics as a feed additive [60]. Higher quantities of short-chain fatty acids can be found in the gut of broiler chickens when probiotics are added to the diet. The pH of digesta will consequently drop. This circumstance favors the growth of helpful bacteria but not of pathogenic bacteria [59]. It can be helpful to comprehend the bacteria that live in the chicken gut in order to help the generation of new probiotic feed additives which can enhance gut health, serve as an alternative to antibiotics and improve chicken welfare in the poultry industry. 

## 5. Conclusions

In conclusion, the inclusion of *P. pentosaceus* GT001 at 4.0 × 10^8^ cfu/g feed as a probiotic in broiler diets significantly increased body weight gain. It improved liver function by lowering triglyceride concentration. Dietary inclusion of *P. pentosaceus* GT001 at 4.0 × 10^8^ cfu/g feed enhanced digestive enzyme production. Antioxidant activities were enhanced and a decreased count of E. coli in the gut was observed. It could therefore be concluded that *Pediococcus pentosaceus* GT001 can be used as a probiotic feed additive in the poultry industry and can serve as a promising alternative to the use of antibiotics. 

## Figures and Tables

**Figure 1 animals-13-03724-f001:**
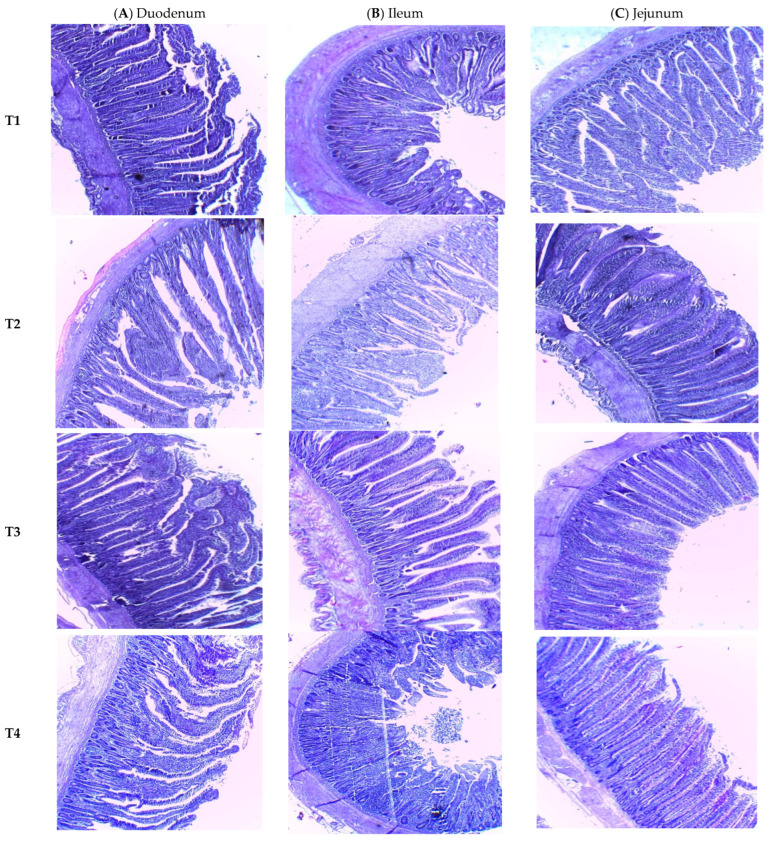
Electron micrograph images of the duodenum (**A**), ileum (**B**) and jejunum (**C**) tissues from the treatment birds. (T1)—basal diet (corn and soybean based); (T2)—basal diet supplemented with antibact 3X 1 g/kg feed; (T3)—basal diet supplemented with *P. pentosaceus* GT001 at 4.0 × 10^8^ cfu/g feed; (T4)—basal diet supplemented with *P. pentosaceus* at 8.0 × 10^8^ cfu/g feed; and (T5)—basal diet supplemented with *P. pentosaceus* at 1.2 × 10^9^ cfu/g.

**Table 1 animals-13-03724-t001:** Composition of basal diet (%).

Ingredients (kg)	Starter (1–21 Days)	Finisher (22–42 Days)
Maize	58.2	63.5
Soya	30.0	25.0
Fish	5.3	5.0
Limestone	1.3	1.3
Soyabean oil	2.0	2.0
L-Lysine	0.2	0.2
DL-Methionine	0.2	0.2
Di calcium phosphate	1.5	1.5
Salt	0.3	0.3
Premix	1.0	1.0
Total	100	100
Calculated composition		
Energy	3251	3255
Crude Protein	22.11	20.08
Total Phosphorus	0.73	0.65
Methionine	0.24	0.23
Methionine + Cysteine	0.93	0.87
Lysine	1.39	1.25
Ether Extract	5.22	5.29
Crude Fibre	3.04	2.85

Premix provided the following per kilogram of diet: 13,000 IU of vitamin A; 1300 IU of vitamin D; 65 IU of vitamin E; 3.4 mg of menadione; 37 mg of pantothenic acid; 6.6 mg of riboflavin; 3.7 mg of folic acid; 39 mg of niacin; 1.0 mg of thiamine; 4.3 mg of vitamin B6; 0.23 mg biotin; 0.075 mg of vitamin B12; 43 mg of choline chloride; 170 mg of zinc; 140 mg of iron; 34 mg of manganese; 16 mg of copper; 0.29 mg of iodine; 0.29 mg of selenium.

**Table 2 animals-13-03724-t002:** Effect of different concentrations of *Pediococcus pentosaceus* GT001 on growth performance of broiler chickens.

Parameters (g)	T1	T2	T3	T4	T5	SEM	*p*-Value
Initial Weight	38.24	38.16	38.05	38.54	38.71	-	-
Final Weight	1468.0 ^a^	1285.9 ^b^	1484.0 ^a^	1421.0 ^ab^	1287.0 ^b^	52.9	0.030
Feed Intake/Day	70.57 ^bc^	63.60 ^d^	75.59 ^a^	71.46 ^b^	67.55 ^c^	1.30	<0.001
Total Weight Gain	1429.8 ^a^	1247.8 ^b^	1446.0 ^a^	1382.5 ^ab^	1248.3 ^b^	52.9	0.030
Average Daily Gain	34.04 ^a^	29.71 ^b^	34.43 ^a^	32.92 ^ab^	29.72 ^b^	1.26	0.030
Feed Conversion Ratio	2.08	2.15	2.20	2.18	2.20	0.05	0.514

T1—basal diet (corn and soybean based); T2—basal diet supplemented with antibact 3X 1 g/kg feed; T3—basal diet supplemented with *P. pentosaceus* GT001 at 4.0 × 10^8^ cfu/g feed; T4—basal diet supplemented with *P. pentosaceus* at 8.0 × 10^8^ cfu/g feed; and T5—basal diet supplemented with *P. pentosaceus* at 1.2 × 10^9^ cfu/g. ^a,b,c,d^ Means in the same row with different superscripts differ significantly (*p* < 0.05). SEM—Standard Error of Mean. (*n* = 10).

**Table 3 animals-13-03724-t003:** Effect of different concentrations of *Pediococcus pentosaceus* GT001 on the liver function of broiler chicken.

Parameters	T1	T2	T3	T4	T5	SEM	*p*-Value
Total Protein (g/L)	28.27	28.42	29.92	27.32	29.77	3.06	0.970
Albumin (g/L)	13.24	17.33	13.25	14.46	14.27	1.44	0.285
Globulin (g/L)	21.02	18.93	20.27	16.85	19.50	2.52	0.808
Creatinine (ummol/L)	35.33 ^a^	22.09 ^b^	20.45 ^b^	15.25 ^b^	25.60 ^ab^	3.08	0.003
AST (U/L)	171.70	141.30	175.5	155.32	145.53	26.20	0.580
ALT (U/L)	12.84 ^b^	6.47 ^c^	15.85 ^ab^	20.13 ^a^	4.81 ^c^	2.59	0.002

^a,b,c^ Means in the same row with different superscripts differ significantly (*p* < 0.05). SEM—Standard Error of Mean. AST—aspartate aminotransferase. ALT—alanine aminotransferase (*n* = 10).

**Table 4 animals-13-03724-t004:** Effect of different concentrations of *Pediococcus pentosaceus* GT001 on lipid profile of broiler chickens.

Parameters (mmol/L)	T1	T2	T3	T4	T5	SEM	*p*-Value
Total cholesterol	299.3 ^a^	116.3 ^b^	281.5 ^a^	212.6 ^ab^	161.7 ^b^	27.7	0.001
Triglycerides	61.41 ^ab^	76.09 ^a^	27.41 ^b^	83.87 ^a^	58.20 ^ab^	8.20	0.001
HDL	57.98 ^ab^	54.74 ^ab^	44.22 ^b^	65.41 ^a^	56.32 ^ab^	4.35	0.040
LDL	200.00 ^ab^	70.50 ^c^	298.80 ^a^	263.00 ^a^	122.9 ^b^	29.30	<0.001

^a,b,c^ Means in the same row with different superscripts differ significantly (*p ˂* 0.05). SEM—Standard Error of Mean. HDL—high density lipoprotein. LDL—low density lipoprotein (*n* = 10).

**Table 5 animals-13-03724-t005:** Effect of different concentrations of *Pediococcus pentosaceus* GT001 on intestinal length and pH of broiler chickens.

Parameters	T1	T2	T3	T4	T5	SEM	*p*-Value
pH							
Duodenum	6.262	6.384	6.208	6.260	6.242	0.123	0.879
Jejunum	6.888 ^a^	6.790 ^a^	6.294 ^b^	6.326 ^b^	6.542 ^ab^	0.156	0.046
Ileum	7.418 ^a^	6.914 ^ab^	6.476 ^b^	6.436 ^b^	6.676 ^b^	0.218	0.028
Length (cm)							
Duodenum	29.20	30.40	30.00	30.80	34.00	1.61	0.300
Jejunum	81.00	74.60	73.60	79.00	84.20	2.92	0.096
Ileum	79.80	76.00	71.80	74.00	69.40	4.90	0.627

^a,b^ Means in the same row with different superscripts differ significantly (*p ˂* 0.05). SEM—Standard Error of Mean. (*n* = 10).

**Table 6 animals-13-03724-t006:** Effect of different concentrations of *Pediococcus pentosaceus* GT001 on the digestive enzymes of broiler chickens.

Parameters (ng/mL)	T1	T2	T3	T4	T5	SEM	*p*-Value
Serum							
Amylase	51.55 ^b^	46.98 ^c^	62.79 ^a^	61.29 ^a^	54.18 ^b^	0.738	<0.001
Lipase	22.63 ^d^	25.94 ^d^	44.82 ^a^	36.36 ^b^	30.51 ^c^	0.867	<0.001
Intestinal							
Amylase	25.13 ^a^	21.88 ^b^	26.52 ^a^	24.36 ^a^	24.34 ^a^	0.536	<0.001
Lipase	12.05 ^c^	10.60 ^d^	15.42 ^a^	13.86 ^b^	13.66 ^b^	0.321	<0.001

^a,b,c,d^ Means in the same row with different superscripts differ significantly (*p* ˂ 0.05). SEM—Standard Error of Mean. (*n* = 10).

**Table 7 animals-13-03724-t007:** Effect of different concentrations of *Pediococcus pentosaceus* GT001 on serum antioxidant capacity of broiler chickens.

Parameters	T1	T2	T3	T4	T5	SEM	*p*-Value
T-AOC (nmol/L)	0.45 ^b^	0.36 ^c^	0.53 ^a^	0.38 ^c^	0.37 ^c^	0.02	<0.001
SOD (nmol/L)	131.48 ^d^	135.24 ^c^	140.99 ^ab^	143.16 ^a^	138.71 ^b^	0.721	<0.001
GHS-Px (U/mL)	564.62 ^c^	579.09 ^c^	615.56 ^b^	633.88 ^a^	623.80 ^ab^	3.78	<0.001
CAT (U/mL)	305.05 ^c^	327.26 ^b^	377.20 ^a^	385.95 ^a^	381.09 ^a^	4.03	<0.001
MDA (U/mL)	4.20 ^a^	3.51 ^bc^	3.90 ^ab^	3.10 ^c^	3.20 ^c^	0.109	<0.001

^a,b,c,d^ Means in the same row with different superscripts differ significantly (*p ˂* 0.05). SEM—Standard Error of Mean. T-AOC—total antioxidant capacity. SOD—superoxide dismutase. GHS-Px—glutathione peroxidase. CAT—catalase. MDA—malondialdehyde (*n* = 10).

**Table 8 animals-13-03724-t008:** Effect of different concentrations of *Pediococcus pentosaceus* GT001 on serum cytokines and immunoglobin of broiler chickens.

Parameters	T1	T2	T3	T4	T5	SEM	*p*-Value
Cytokines							
TNF-α (pg/mL)	116.83 ^a^	104.16 ^b^	105.22 ^b^	93.52 ^c^	100.08 ^cb^	2.15	<0.001
IL 6 (pg/mL)	63.24 ^a^	57.74 b^c^	62.92 ^a^	55.67 ^c^	59.45 ^b^	0.781	<0.001
IL 10 (pg/mL)	29.92 ^b^	31.01 ^b^	34.28 ^a^	35.05 ^a^	33.36 ^a^	0.510	<0.001
Immunoglobin							
IgA (g/L)	0.92 ^bc^	0.87 ^c^	1.06 ^a^	1.01 ^ab^	0.93 ^bc^	0.0244	<0.001
IgG (g/L)	8.17 ^b^	8.08 ^b^	8.69 ^a^	8.76 ^a^	8.67 ^a^	0.0898	<0.001
IgM (g/L)	0.78 ^b^	0.77 ^b^	0.94 ^a^	0.90 ^a^	0.87 ^a^	0.0191	<0.001

^a,b,c^ Means in the same row with different superscripts differ significantly (*p* < 0.05). SEM—Standard Error of Mean. TNF-α—tumor necrosis factor-alpha. IL 6—interleukin 6. IL 10—interleukin 10. IgA—Immunoglobulin A. IgG—Immunoglobulin G. IgM—Immunoglobulin M (*n* = 10).

**Table 9 animals-13-03724-t009:** Effect of different concentrations of *Pediococcus pentosaceus* GT001 on the small intestinal morphology in broiler chickens.

Parameters (μm)	T1	T2	T3	T4	T5	SEM	*p*-Value
Duodenum							
Villus height	1248.60 ^b^	1090.60 ^c^	1377.00 ^a^	1302.00 ^ab^	1212.20 ^b^	28.2	<0.001
Crypt depth	219.00 ^ab^	217.60 ^ab^	229.00 ^ab^	232.00 ^a^	208.00 ^b^	5.06	0.023
VH/CP	5.72 ^ab^	5.02 ^b^	6.04 ^a^	5.63 ^ab^	5.83 ^a^	0.18	0.010
Ileum							
Villus height	960.60 ^a^	866.20 ^b^	989.20 ^a^	938.80 ^ab^	951.80 ^a^	18.50	0.002
Crypt depth	207.40 ^a^	184.20 ^b^	158.00 ^c^	186.40 ^b^	186.00 ^b^	3.96	<0.001
VH/CP	4.64 ^b^	4.70 ^b^	6.27 ^a^	5.06 ^b^	5.12 ^b^	0.141	<0.001
Jejunum							
Villus height	1179.40 ^b^	1221.40 ^ab^	1312.20 ^a^	1237.60 ^ab^	1209.60 ^ab^	25.4	0.018
Crypt depth	229.00 ^a^	221.60 ^ab^	204.60 ^b^	216.00 ^ab^	229.20 ^a^	4.11	0.002
VH/CP	5.16 ^c^	5.52 ^bc^	6.43 ^a^	5.73 ^bc^	5.28 ^b^	0.130	<0.001

^a,b,c^ Means in the same row with different superscripts differ significantly (*p* < 0.05). SEM—Standard Error of MeanVH/CP—Villus height and crypt depth ratio (*n* = 10).

**Table 10 animals-13-03724-t010:** Effect of different concentrations of *Pediococcus pentosaceus* GT001 on duodenal microbial count in broiler chickens expressed as log_10_.

Treatments	*Pediococcus* spp. (10^3^)	Total Viable Count (TVC) (10^6^)	Total Coliform Count (TCC) (10^2^)	Non-Lactose Fermenters (NLF)	*E. Coli* (10^1^)	*Enterobacter* Count (EBC)	*Enterococcus* Count (ECC)	*Salmonella* Count (SC)
T1	0.0013 ^b^	1.46 ^ab^	1.21 ^a^	No growth	0.15	No growth	No growth	No growth
T2	0.0012 ^b^	2.38 ^a^	0.013 ^a^	No growth	0.22	No growth	No growth	No growth
T3	2.32 ^ab^	5.22 ^b^	0.0002 ^b^	No growth	0.13	No growth	No growth	No growth
T4	4.7 ^a^	1.26 ^ab^	0.0008 ^b^	No growth	0.35	No growth	No growth	No growth
T5	3.76 ^a^	2.45 ^a^	0.0004 ^b^	No growth	0.12	No growth	No growth	No growth
SEM	447	24.3	4.94	-	0.753	-	-	-
*p*-Value	0.002	0.011	<0.001	-	0.306	-	-	-

**^a,b^** Means in the same column with different superscripts differ significantly (*p* < 0.05). SEM—Standard Error of Mean. (*n* = 10).

**Table 11 animals-13-03724-t011:** Effect of different concentrations of *Pediococcus pentosaceus* GT001 on jejunum microbial count in broiler chickens expressed as log_10_.

Treatments	*Pediococcus* spp. (10^3^)	Total Viable Count (TVC) (10^6^)	Total Coliform Count (TCC) (10^2^)	Non-Lactose Fermenters (NLF)	*E. Coli* (10^1^)	*Enterobacter* Count (EBC)	*Enterococcus* Count (ECC) (10^1^)	*Salmonella* Count (SC)
T1	0.0012 ^b^	6.81	5.55 ^a^	No growth	1.71 ^a^	No growth	0.57 ^c^	No growth
T2	0.0017 ^b^	6.06	4.82 ^a^	No growth	1.08 ^b^	No growth	1.89 ^a^	No growth
T3	1.56 ^ab^	1.91	1.67 ^b^	No growth	1.12 ^b^	No growth	1.13 ^b^	No growth
T4	5.89 ^a^	4.11	1.26 ^b^	No growth	4.20 ^b^	No growth	1.04 ^bc^	No growth
T5	3.76 ^a^	4.62	0.24 ^c^	No growth	1.20 ^b^	No growth	0.42 ^d^	No growth
SEM	382	1.42	63.6	-	4.27	-	0.881	-
*p*-Value	<0.001	0.269	0.0040	-	0.015	-	<0.001	-

^a,b,c,d^ Means in the same column with different superscripts differ significantly (*p* < 0.05). SEM—Standard Error of Mean. (*n* = 10).

**Table 12 animals-13-03724-t012:** Effect of different concentrations of *Pediococcus pentosaceus* GT001 on ileum microbial count in broiler chickens expressed as log_10_.

Treatments	*Pediococcus* spp. (10^3^)	Total Viable Count (TVC) (10^9^)	Total Coliform Count (TCC) (10^4^)	Non-Lactose Fermenters (NLF) (10^1^)	*E. Coli* (10^2^)	*Enterobacter* Count (EBC) (10^1^)	*Enterococcus* Count (ECC) (10^1^)	*Salmonella* Count (SC)
T1	0.0062 ^c^	4.78 ^a^	8.67 ^a^	2.40	7.31 ^a^	1.61	2.04 ^a^	No growth
T2	0.0065 ^c^	1.16 ^b^	5.51 ^ab^	1.83	7.96 ^a^	2.63	1.82 ^a^	No growth
T3	2.81 ^b^	0.14 ^b^	1.82 ^b^	1.60	5.22 ^b^	2.41	1.74 ^a^	No growth
T4	4.10 ^a^	0.94 ^b^	2.13 ^b^	4.43	1.73 ^bc^	1.15	1.54 ^ab^	No growth
T5	4.31 ^a^	1.03 ^b^	0.57 ^b^	5.28	1.13 ^c^	1.16	1.17 ^b^	No growth
SEM	159	292	10.85	8.48	64.4	3.13	0.96	-
*p*-Value	<0.001	0.001	0.015	0.088	0.002	0.074	0.015	-

^a,b,c^ Means in the same column with different superscripts differ significantly (*p <* 0.05). SEM—Standard Error of Mean. (*n* = 10).

## Data Availability

The corresponding author may provide the data that back up this study’s conclusions upon request.

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
