# Peer review of "Different Concentrations of Probiotic Pediococcus pentosaceus GT001 on Growth Performance, Antioxidant Capacity, Immune Function, Intestinal Microflora and Histomorphology of Broiler Chickens"

_animals, 2023, doi:10.3390/ani13233724_

Round 1

Reviewer 1 Report

Comments and Suggestions for Authors

Comments on the Quality of English Language

The manuscript is well-written and minor editing of English language is required. 

Author Response

General comment: The manuscript tilted “Different concentration of probiotic Pediococcus pentosaceus GT001 on growth performance, antioxidant capacity, immune function, intestinal microflora and histomorphology of broiler chickens” is well-written and reports research with some originality. However, the following comments should be addressed:

Authors response: We appreciate the reviewer's insightful remarks and recommendations. We have implemented extensive changes based on the recommendations. A revised manuscript with the detailed modifications have been highlighted

Comment 1: Lines 37,105,122: Avoid using Arabic numerals at the beginning of sentences.

Authors’ response: According to your suggestion, the sentences starting with Arabic numerals have been modified. (Please see lines 39, 119 and 136 in the revised version. It’s been highlighted in red).

       Comment 2: The basal diet is in powder form or pellet form?

Authors’ response. The basal diet was in a mashed form. (Please see line 130 in the revised version. It’s been highlighted in red).

Comment 3: Lack of information for antibact 3X.

Authors’ response: Information on antibact 3X has been included. (Please see lines 139 and 140 in the revised version. It’s been highlighted in red).

Comment 4: Effect of different concentrations of …

Authors’ response: The grammatical error has been corrected. (Please see line 258 in the revised version. It’s been highlighted in red).

Comment 5: The FCR was quite high for Ross 708 broilers. How do you explain?

Authors’ response: Please there has been an explanation why FCR seem to be quite high in Ross 708 broilers.  (Please see lines 566-574 in the revised version. It’s been highlighted in red).

Comment 6: Line224-236: revise the p values. For example, in line 236, the p=0.514.

Authors’ response: Please all p values have been revised. (Kindly see lines 271, 272, 274, 276, 277, 278, 319, 330, 331, 332, 333, 352,353, 354, 356, 357, 358, 360, 361, 362, 377, 378, 379, 382, 383, 384, 385, 386, 401, 402, 403, 406, 407, 420, 421, 422, 424, 425, 426, 428, 429,431, 432 and 434 in the revised version. They have all been highlighted in red).

Comment 7: In the whole manuscript, revise “different concentration” to “different concentrations”

Authors’ response: Thank you. Different concentration has been modified to different concentrations in the manuscript as suggested. (Kindly see lines 280, 300, 320, 342, 365 and 388 in the revised version. They have all been highlighted in red).

Comment 8: Table 3: The units for AST and ASL: U/L

Authors’ response: The units of AST and ASL have been changed from u/l to U/L. Thank you. (Kindly see table 3 in the revised version. It’s been highlighted in red).

Comment 9: Table notes for table 2,6,7,11: a,b,c,d Means in the same row with different superscripts differ significantly (p˂ 0.05)

Authors’ response: Table notes for table 2,6,7 and 11 have be modified. Thank you. (Kindly see lines 263, 347, 370 and 509 in the revised version. It’s been highlighted in red).

Comment 10: Table note for table 5: a, b Means in the same row with different superscripts differ significantly (p Ë‚ 0.05)

Authors’ response: Table note for table 5 has be modified. Thank you. (Kindly see lines 325and 489 in the revised version. It’s been highlighted in red).

Comment 11: Table 7: Since the activities of antioxidant enzymes were increased, why the T-AOC was decreased in GT001-treated groups?

Authors’ response: Why the T-AOC was decreased in GT001-treated groups has be explained. (Kindly see lines 689-693 in the revised version. It’s been highlighted in red).

Comment 12: Figure1 was not cited in the manuscript.

Authors’ response: Thank you. Figure 1 is just an image showing us how the small intestine looked like among the treatment groups.

Comment 13: 468: ileum

Authors’ response: it has been corrected. Thank you. (Kindly see line 537 in the revised version. It’s been highlighted in red). 

Comment 14: Lines 500-504: These discussions on triglyceride should be removed to 4.3. Besides, the alterations of creatinine and ALT were not discussed properly in 4.2.

Authors’ response: The modification has been done as suggested. (Kindly see lines 580-598 and 606-610 in the revised version. It’s been highlighted in red). 

Comment 15: Why the lower GT001 dosage (4.0 x 108 cfu/g) had better effects than the higher dosages? According to the results, 1.2 x 109 cfu/g can even cause adverse effect on the parameters (such as decreased final weight).

Authors’ response: Thank you. The reason has been explained. (Kindly see lines 563-565 in the revised version. It’s been highlighted in red). 

Reviewer 2 Report

Comments and Suggestions for Authors

To improve the quality of publication, I suggest the following:

1. The methodology must clarify what equipment was used to conduct the research. Indicate the brand of the device for determining biochemical blood parameters, pH.

2. In the methodology, clarify what sample of birds was used in the study of biochemical parameters, histological studies, and studies of intestinal contents. How many heads from the group were subjected to control slaughter?

3. In the description of statistical methods, indicate the method used to find the difference between the study groups.

4. Indicate the sample sizes In Tables 2-12.

5. There is a comment about of the groups for research After each table. It is enough to leave this after Table 2 and not present it further.

6. Lines 489-490 indicate the concentration of Ca and Z in the blood serum. The authors did not determine these indicators. It's not worth mentioning.

7. Line 544 – a typo in the source number. You need to write 29 instead of 39.

8. No reference to source 46 was found in the text.

9. I would like an explanation from the authors why the level of triglycerides in the blood serum decreases in groups T3 and T5.

Author Response

General comment: To improve the quality of publication, I suggest the following:

Authors response: We appreciate the reviewer's insightful remarks and recommendations. We have implemented extensive changes based on the recommendations. A revised manuscript with the detailed modifications have been highlighted.

Comment 1: The methodology must clarify what equipment was used to conduct the research. Indicate the brand of the device for determining biochemical blood parameters, pH.

Authors’ response: Thank you for the input. The brand of device used for the determination of biochemical blood parameters and pH have been stated I the methodology. (Kindly see line 175 and 199 in the revised version. It’s been highlighted in red). 

Comment 2: In the methodology, clarify what sample of birds was used in the study of biochemical parameters, histological studies, and studies of intestinal contents. How many heads from the group were subjected to control slaughter?

Authors’ response: The sampled number of birds used has be stated. Thank you. (Kindly see line 170 and 197 in the revised version. It’s been highlighted red). 

Comment 3: In the description of statistical methods, indicate the method used to find the difference between the study groups.

Authors’ response: The method used in finding the difference among the treatment groups has been stated. (Kindly see line 248 and 249 in the revised version. It’s been highlighted in red). 

Comment 4: Indicate the sample sizes in Tables 2-12.

Authors’ response: The sample sizes have been indicated in table 2-12. Thank you. (Kindly see lines 264, 287, 307, 326, 348, 372, 395, 415, 490, 510 and 535 in the revised version. It’s been highlighted in red). 

Comment 5: There is a comment about the groups for research after each table. It is enough to leave this after Table 2 and not present it further.

Authors’ response: As suggested, the comment about treatment groups is only under table 2. Thank you. (Kindly see lines 282-284, 302-304, 322-324, 344-346, 367-369, 390-392, 411-413, 486-488, 506-508 and 531-533 in the revised version. It’s been highlighted in red). 

Comment 6: Lines 489-490 indicate the concentration of Ca and Z in the blood serum. The authors did not determine these indicators. It's not worth mentioning.

Authors’ response: Comment on Ca and P has been deleted as suggested. Thank you. (Kindly see line 560-561 In the revised version. It’s been highlighted in red).   

Comment 7: Line 544 – a typo in the source number. You need to write 29 instead of 39.

Authors’ response: It has been modified. Thank you. (There has been a general modification of citation in the revised version of the manuscript. They have all been highlighted red).  

Comment 8: No reference to source 46 was found in the text.

Authors’ response: It has been modified. Thank you. (There has been a general modification of citation in the revised version of the manuscript. They have all been highlighted red).  

Comment 9: I would like an explanation from the authors why the level of triglycerides in the blood serum decreases in groups T3 and T5.

Authors’ response: Why triglycerides level in the blood decreased in T3 and T5 has been explained in the manuscript as requested. (Kindly see lines 611-614 in the revised version. It’s been highlighted in red).   

Round 2

Reviewer 1 Report

Comments and Suggestions for Authors

In the revised manuscript, the authors addressed the issues and it is recommended for publication.